# Integrated analysis of metabolome and microbiome in a rat model of perimenopausal syndrome

Yanqiu Wei,[1] Juanjuan Shi,[1] Jianhua Wang,[2] Zongyan Hu,[3] Min Wang,[4] Wen Wang,[1] Xiujuan Cui[1]

**ABSTRACT** The objectives of this study are to examine the disparities in serum and intestinal tissue metabolites between a perimenopausal rat model and control rats and to analyze the diversity and functionality of intestinal microorganisms to determine the potential correlation between intestinal flora and metabolites. We established a rat model of perimenopausal syndrome (PMS) and performed an integrated analysis of metabolome and microbiome. Orthogonal partial least-squares discriminant analysis scores and replacement tests indicated distinct separations of anion and cation levels between serum and intestinal samples of the model and control groups. Furthermore, lipids and lipid-like molecules constituted the largest percentage of HMDB compounds in both serum and intestinal tissues, followed by organic acids and derivatives, and organoheterocyclic compounds, with other compounds showing significant variability. Moreover, analysis of diversity and functional enrichment of the intestinal microflora and correlation analysis with metabolites revealed significant variability in the composition of the intestinal flora between the normal control and perimenopausal groups, with these differentially expressed intestinal flora strongly correlated with their metabolites. The findings of this study are expected to contribute to understanding the indications and contraindications for estrogen application in perimenopausal women and to aid in the development of appropriate therapeutic agents.

**IMPORTANCE** In this work, we employed 16S ribosomal RNA gene sequencing to analyze the gut microbes in stool samples. In addition, we conducted an ultra-high-performance liquid chromatography-tandem mass spectrometry-based metabolomics approach on gut tissue and serum obtained from rats with perimenopausal syndrome (PMS) and healthy controls. By characterizing the composition and metabolomic properties of gut microbes in PMS rats, we aim to enhance our understanding of their role in women's health, emphasizing the significance of regulating gut microbes in the context of menopausal women's well-being. We aim to provide a theoretical basis for the prevention and treatment of PMS in terms of gut microflora as well as metabolism.

**KEYWORDS** perimenopausal syndrome, gut microbes, 16S ribosomal RNA gene sequencing, metabolic profiles, ultra-high-performance liquid chromatography-tandem mass spectrometry

During the menopause transition, women typically experience a gradual alteration in ovarian activity and a physiological decline in the functioning of the hypo-thalamic-pituitary-ovarian axis, leading to fluctuating hormone levels that contribute to the development of perimenopausal syndrome (PMS) (1). Symptoms including hot flashes, night sweats, sleep disturbances, sexual dysfunction, and mood disorders pose significant challenges to women experiencing this syndrome (2). Menopause is characterized by the total cessation of menstrual periods for 12 consecutive months, while perimenopause, often referred to as the menopausal transition, represents the

**Peer Reviewer** Marisol I. Dothard, Boston University, Boston, Massachusetts, USA

Address correspondence to Wen Wang, wenwenpeking@126.com, or Xiujuan Cui, drcuixj@163.com.

The authors declare no conflict of interest.

See the funding table on p. 15.

phase during which women move from their reproductive years (premenopause) into menopause (3, 4). PMS not only detrimentally impacts individual quality of life but also potentially links to the development of diabetes, osteoporosis, and breast cancer (5–8). Consequently, the prevention and treatment of PMS demand more attention.

Increasing evidence demonstrates that gut microbes are closely related to health and disease. Gut microbes are often regarded as a "virtual" organ within the human body. The intestine harbors approximately $10^{14}$ bacteria, predominantly residing in the lumen and in the mucus (9, 10). Gut dysbiosis is associated with various female reproductive and endocrine diseases, including endometriosis (11), polycystic ovary syndrome (12), obesity, and sexual precocious puberty (13). Common in premenopausal women, endometriosis may be mediated by high estrogen levels (14). During menopause and postmenopause, the depletion of circulating estrogen can lead to various negative health outcomes. There is a variation in gut microbial composition between premenopausal and postmenopausal women (15). A study demonstrated that the level of estrogen metabolites in urine is positively correlated with the diversity of the gut microbiota (16). Osteoporosis results from continuous hypoestrogenemia. Research has found that the gut microbiota interfere with hormone secretion, estrogen levels, metabolism, and immune function, all of which influence bone metabolism (17, 18). Numerous menopause-related symptoms and signs are lack of estrogen production. However, the relationship between gut microbiota and PMS remains poorly characterized. Consequently, the association between gut dysbiosis and PMS is not well understood. Furthermore, it remains to be explored how gut microbes are related to PMS metabolism and which specific metabolites they affect in PMS.

Recently, metabolomics has emerged as a promising technology for quantitatively measuring dynamic metabolomic profiles in living subjects under specific pathophysiological conditions. The commonly used methods for sample analysis include mass spectrometry (MS) and 1 H nuclear magnetic resonance (NMR). A key advantage of the metabolomics approach is its ability to simultaneously measure numerous small molecular metabolites (19, 20). It is particularly effective for diseases characterized by complex metabolic responses (20, 21). Consequently, it is extensively utilized in early diagnosis, the discovery of novel biomarkers, the identification of treatment targets, and the elucidation of disease mechanisms, especially in metabolic diseases. Lv et al. link metabolic changes during the PMS with ferroptosis, suggesting that GNS might regulate ferroptosis by modulating the metabolic pathways of organic acids and their derivatives, as well as lipids and lipid-like molecules (22). However, there is still a lack of research on a comprehensive analysis of perimenopausal metabolism and 16srRNA.

To investigate the taxonomic classification of gut microbes and the associated metabolites in gut tissue and serum of perimenopausal women, we utilized 16S ribosomal RNA (16S rRNA) gene sequencing to analyze gut microbes in stool samples. Furthermore, we conducted ultra-high-performance liquid chromatography-tandem mass spectrometry (UHPLC-MS/MS)-based metabolomics approach on gut tissue and serum from rats with PMS and healthy controls. By characterizing the composition and metabolomic properties of gut microbes in PMS rats, we aim to enhance our understanding of their role in women's health, highlighting the importance of regulating gut microbes in the context of perimenopausal women's well-being. Our research seeks to establish a theoretical basis for the prevention and treatment of PMS, focusing on gut microflora and metabolism.

## MATERIALS AND METHODS

### Animals and treatment

Sixteen female Sprague Dawley rats (SPF grade, 12 weeks old, 300–320 g) were obtained from Vital River Laboratory Animal Technology (Pinghu, China). All animals were maintained on a 12-h light/dark cycle at a constant temperature (22 ± 2℃) with free access to food and water. One week later, eight rats were randomly selected to

form the sham group, while the remaining rats underwent ovariectomy to establish the perimenopausal model (OVX model group). Following an intraperitoneal injection of 2% pentobarbital sodium (0.3 mL/100 g), bilateral oophorectomy was performed. Longitudinal incisions (about 2 cm) were made at the length of 1/3 the trunk, 1–2 cm from each side of the spine, and the adipose tissue was extracted with forceps to locate the ovaries (flesh-colored, thin thread-like, irregular tissue) attached to the uterine horns. After ligation, the ovaries were excised, and the incisions were sutured. For the sham group, similar procedures were performed but the ovaries were not removed. After the operation, the animals were carefully reared. For the sham group, similar procedures were conducted without removing the ovaries.

## Vaginal smear observation

The estrous cycle of the rats was monitored before the operation and again on the fifth-day post-operation. During the estrous cycle, vaginal exfoliated cells were collected from the rats once daily at noon and once at night. The vaginal opening was exposed, and a saline-dipped cotton swab was gently inserted and rotated in a consistent direction. The cotton swab was coated with the exfoliated vaginal epithelial cells, was then smeared onto a slide, and the cells were stained with methylene blue solution. After staining for 15 minutes, the cells were washed with phosphate-buffered saline, dried, and observed under an optical microscope (OLYMPUS, Tokyo, Japan; scale bar = 20 µM).

## Measurement of FSH, luteotropic hormone, and female hormone estradiol levels

After the vaginal smear observation, the rats were anesthetized with an intraperitoneal injection of 2% pentobarbital sodium, and blood samples were obtained from the abdominal aorta using tubes without heparin to separate serum. The serum levels of follicle-stimulating hormone (FSH), luteotropic hormone (LH), and E2 were measured using ELISA with the corresponding commercial kits (Calvin Biotechnology, Suzhou, China).

## Histology examination

The rats in different groups were euthanatized through intraperitoneal injection of the overdose of pentobarbital sodium. The vaginal tissues were fixed, dehydrated, embedded in paraffin, and then cut into 4 µm thickness slices. After being dewaxed and rehydrated, the sections were executed for Masson's trichrome staining assay (Maxim Biotech, Fuzhou, China). Images were captured using an optical microscope (scale bar = 100 µM).

## Western blotting

The vaginal tissues were lysed using RIPA buffer (BOSTER, Wuhan, China), and then protein concentrations were then determined using a BCA Protein Kit (Thermo Fisher Scientific, Rockford, MD, USA). This was followed by the separation of the protein products *via* SDS polyacrylamide gel electrophoresis and their subsequent transfer onto PVDF membranes. We used 5% nonfat milk to block the membranes. Afterward, the primary antibodies (α-SMA, collagen I, collagen II, and GAPDH; Abcam, Cambridge, UK; all diluted 1:1,000) were applied to the membranes and incubated overnight at 4℃, followed by the addition of the corresponding secondary antibody (Abcam; dilution 1:3,000) for 1 hour at room temperature. Immunoblotting results were visualized using an ECL detection kit (Tanon, Shanghai, China).

## Sample preparation

The serum samples were prepared using the method described in a previous study (23). A 400 mL extraction solvent comprising methanol and acetonitrile in a 1:1 ratio (vol/vol)

was added to a 100 mL sample. The mixture was vortexed for 30 seconds and subsequently sonicated for 10 minutes in an ice-water bath. The solution was then incubated at −20°C for 1 hour and centrifuged at 4°C and 12,000 rpm for 15 minutes to precipitate the proteins, to prevent the obstruction of macromolecules in the chromatographic column and electrospray ionization probe. The supernatant (420 μL) was transferred into EP tubes and subjected to vacuum concentration without heating. The concentrated extracts were then added to a 100 mL extraction solution (acetonitrile:water = 1:1, vol/vol), followed by vortexing for 30 seconds and sonication for 10 minutes in a water bath at 4°C. After centrifugation at 12,000 rpm and 4°C for 15 minutes, the resulting mixture was obtained. The supernatant was transferred into a fresh 2 mL glass for subsequent detection.

Intestinal tissues (50 mg) were placed in a 2-mL centrifuge tube, followed by the addition of a grinding bead with a diameter of 6 mm. For metabolite extraction, 400 μL of an extraction solution consisting of methanol and water (4:1, vol/vol) was used, including an internal standard (L-2-chlorophenylalanine) at a concentration of 0.02 mg/mL. The samples were ground using a frozen tissue grinder (Wonbio-96c; Wanbo Biotechnology, Shanghai, China) for 6 minutes at −10°C at a frequency of 50 Hz. Subsequently, low-temperature ultrasonic extraction was performed for 30 minutes at 5°C at a frequency of 40 kHz. After the samples were at −20°C for 30 minutes, they were centrifuged for 15 minutes at 4°C at a speed of 13,000 g. The resulting supernatant was subsequently transferred to the injection vial for UHPLC-MS/MS analysis. To ensure system conditioning and quality control, a pooled quality control sample was prepared by combining equal volumes of all samples.

## UHPLC-MS/MS-based metabolomics analysis

The sample was analyzed using a Thermo UHPLC-Q Exactive HF-X system equipped with an ACQUITY HSS T3 column (100 mm × 2.1 mm ×1.8 μm; Waters, Milford, MA, USA). The flow rate was set at 0.40 mL/min, and the column temperature was maintained at 40°C. For the mobile phases, we employed a mixture of water and acetonitrile (95:5, vol/vol) containing 0.1% formic acid (solvent A), and a combination of acetonitrile, isopropanol, and water (47.5:47.5, vol/vol) with 0.1% formic acid (solvent B). In positive ion mode, separation gradient conditions were set as follows: starting from 0%, solvent B increased to 20% within the first 3 minutes; it then further increased to 35% between 3 and 4.5 minutes; reaching complete replacement by solvent B from 5 to 6.3 minutes; this was followed by a return to initial conditions over a period of one-tenth of a minute, culminating in a total runtime of 8 minutes. In the negative ion mode, separation gradient conditions were adjusted accordingly: starting from 0%, solvent B gradually increased to 5% within 1.5 minutes; it then continued increasing to 10% over 2 minutes; next, it increased up to 30% between 2 and 4.5 minutes; finally, reaching full replacement by solvent B at the 5-minute mark until 6.3 minutes, before reverting completely in just under two-thirds of a minute, for a total runtime of 8 minutes.

The Thermo UHPLC-Q Exactive HF-X Mass Spectrometer was used for MS analysis. It features an electrospray ionization source operating in both positive and negative modes. The optimal experimental conditions included the following: source temperature was set at 425°C; ion-spray voltage was set at 3,500 V in positive mode and −3,500 V in negative mode; sheath gas flow rate was maintained at 50 arb; Aux gas flow rate was set to 13 arb; and rolling collision energy of 20–40–60 V was applied for MS/MS analysis. The resolution for MS/MS measurements reached 7,500, while full MS achieved a resolution of 60,000. Data acquisition followed a Data Dependent Acquisition mode within a mass range spanning from 70 to 1,050 m/z.

## Metabolome data and pathway enrichment analysis

The Progenesis QI software (Waters Corporation, Milford, USA) was utilized to preprocess the raw data obtained from UHPLC-MS/MS analysis. The resulting preprocessed data matrix in CSV format included information on metabolite names, sample details,

and mass spectral response intensity. To ensure accuracy, internal standard peaks and false-positive peaks were removed based on the pooled peak data and data matrix. Metabolite identification was performed using Majorbio Database, Metlin (https://metlin.scripps.edu/), and HMDB (http://www.hmdb.ca/) databases.

The multivariate data analysis was an orthogonal partial least-squares discriminant analysis (OPLS-DA). The unsupervised PCA was used to illustrate the distribution of origin data and overall separation. The supervised OPLS-DA, on the other hand, was performed to obtain maximal covariance between the measured data and the response variable and validated using sevenfold cross-validation and 200 permutation tests. The validity of the OPLS-DA model was evaluated using R2Y and Q2, which were parameters for model stability and the ability to explain and predict the raw data. R2Y and Q2 values are closer to 1, suggesting a better model (24). Compounds with variable importance in projection (VIP) values ≥ 1.0, $P$ value < 0.05, and |log2(Foldchange)| ≥0 were considered as potential differential expressed metabolites between control and sepsis samples. The results of screening differential metabolites were visualized using a volcano plot. We searched the Kyoto Encyclopedia of Genes and Genomes (KEGG, http://www.genome.jp/kegg) (25) and MetaboAnalyst (http://www.metaboanalyst.ca/) (26) to explore the key metabolic pathways represented by the differential metabolites identified by the above experiments.

## 16S rRNA sequencing of gut microbiota

The HiPure Bacterial DNA Kit, provided by Megan (China), was used for extracting bacterial genomic DNA. The quality and quantity of the extracted DNA were assessed using agarose gel electrophoresis and Qubit analysis, respectively (Thermo Fisher Scientific).

After premixing with NEBnext-Ultra-II-Q5-MasterMix, the V3-V4 region of the 16S rRNA gene was amplified (NEB, USA). The final library product was evaluated using the Agilent 2200 Tape Station and Qubit (Life Technologies, USA). The first batch of samples underwent pair-end 250 bp sequencing on the Miseq platform (Illumina, USA), while the second batch was sequenced on the NovaSeq platform (Illumina, USA) with pair-end 250 bp. According to the RS_ReadsOfinsert1 protocol, we obtained raw data. We utilized the QIIME package (Version 1.9.1, http://qiime.org/install/index.html) to obtain high-quality sequences. For operational taxonomic units (OTUs) cluster analysis with a similarity cutoff of 97%, UPARSE (version 7.1) was employed. Chimeric sequences were detected and eliminated. To perform species taxonomic analyses against the Silva 16S rRNA database, we used the Ribosomal Database Program classifier Bayesian algorithm with a confidence threshold of 70%. Subsequently, beta diversity analysis and community composition analysis were conducted.

## Statistical analysis

The variations among the data were gauged by applying Student's $t$-tests. Wilcoxon rank-sum tests were conducted to compare the continuous variables between groups. Fisher's exact test was used to compare categorical variables between groups. Spearman's rank test was used for the correlation analysis. Data analysis was performed in SPSS software v22.0. The data were indicated as the mean ± standard deviation (SD). $P$-values < 0.05 were considered statistically significant.

## RESULTS

### Establishment of perimenopausal rat model

The characteristics of changes in vaginal smear cells of rats during estrus cycle are as follows: (i) proestrus: predominantly nucleated epithelial cells with few keratinocytes; (ii) estrus: a large area of keratinocytes with few nucleated epithelial cells; (iii) metestrus: nucleated epithelial cells, keratinocytes, and leukocytes were uniformly mixed; and (iv) diestrus: primarily leukocytes with small amounts of mucus. Before surgery, we found

that rats in the control and model groups showed a regular estrous cycle, while after surgery, the estrous cycle of rats in the model groups was disturbed (Fig. 1A). Compared with the control group, the serum FSH and LH levels in the model group were significantly increased (Fig. 1B, $P < 0.001$), whereas E2 level was reduced remarkably ($P < 0.001$). Masson's trichrome staining results showed that compared with the control group, the epithelia of the vaginal wall of rats in the model group were atrophied. At the same time, the smooth muscle layer of vaginal wall of rats also changed obviously, with smaller smooth muscle bundles and disordered arrangement (Fig. 1C). Furthermore, we also determined the protein levels of collagen I, collagen II, and α-SMA in vaginal tissues by western blotting, and the results showed that the protein level of collagen I was increased ($P < 0.01$) and the protein levels of collagen II and α-SMA were decreased ($P < 0.01$) in the model group compared to the control group (Fig. 1D and E). All these results implied that the perimenopausal rat model was established successfully.

## Multivariate data analysis of serum and intestinal metabolites

Figure 2 displays the results of the analysis of OPLS-DA scores and replacement tests. The OPLS-DA model was constructed to discriminate the differential metabolites between the two groups. In this model, both serum (Fig. 2A) and intestinal (Fig. 2C) samples present distinct separations of anion and cation levels between the model and control groups. The corresponding replacement test results (Fig. 2B and D) further demonstrate that the OPLS-DA model exhibited no overfitting. The parameter scores of OPLS-DA are shown in Table 1. In serum and intestinal samples, the proportion of variance explained by the OPLS-DA model ($R^2Y$) was 97.9% and 88.7%. The cumulative values of $Q^2$ in the serum and intestinal samples (0.778, 0.681) were all above 0.50, which illustrated the stability and reliability of the model. The good quality and reliability of this OPLS-DA model were confirmed, meaning it was suitable to explore the differences between the two groups in this study. Subsequently, this allowed for the identification of metabolites in the two groups.

## Differential metabolite expression profiling

The volcano plots (Fig. 3) were used to visualize metabolites that exhibited a statistically significant difference. At the cation level, 1,623 metabolites were detected, of which 205 increased and 55 decreased (Fig. 3A). At the anion level, 1,377 metabolites were identified, with 47 increasing and 38 decreasing (Fig. 3B). To cluster the differential metabolites, a complete linkage method was employed, resulting in the formation of a heat map (Fig. 3C and D) that incorporates the expression profile and VIP of the metabolites. The heat map reveals that, at the cation level, metabolites such as tetrahydrocorticosterone, Ser Leu Ile, isoline, concanamycin a, benzoylhypaconine, CDP-DG, and 5-HETE, etc. exhibited a decrease in the majority of the model groups, while an increase was observed in the control groups. Conversely, metabolites such as DG[2:0/20:3(5Z,11Z,14Z)-O (8,9)/0:0], gentiobioside, 2,5-anhydro-D-mannose, (R)−3-hydroxybutyrylcamitine, phenoxomethylpenicilloyl, gibberellin A51-catabolite, and RHODAMINE 6G displayed an increase in the model groups and a decrease in most of the control groups. Notably, all metabolites listed in Fig. 3C and D exhibited statistically significant differences between the two groups.

## Metabolite classification and KEGG functional enrichment analysis

Figure 4 presents the classification results of HMDB compounds in serum and intestinal tissue samples. The results showed that the proportion of lipid and lipid-like molecules was the highest (81, 33.41%), followed by organic acids and their derivatives (60, 24.79%), organoheterocyclic compounds (32, 13.22%), organic oxygen compounds (23, 9.5%), and benzenoids (13, 5.37%) in serum samples (a total of 161 compounds) (Fig. 4A). In intestinal tissue samples (a total of 78 compounds), the classification with the highest percentage also is lipids and lipid-like molecules (37, 47.44%), followed by organic acids

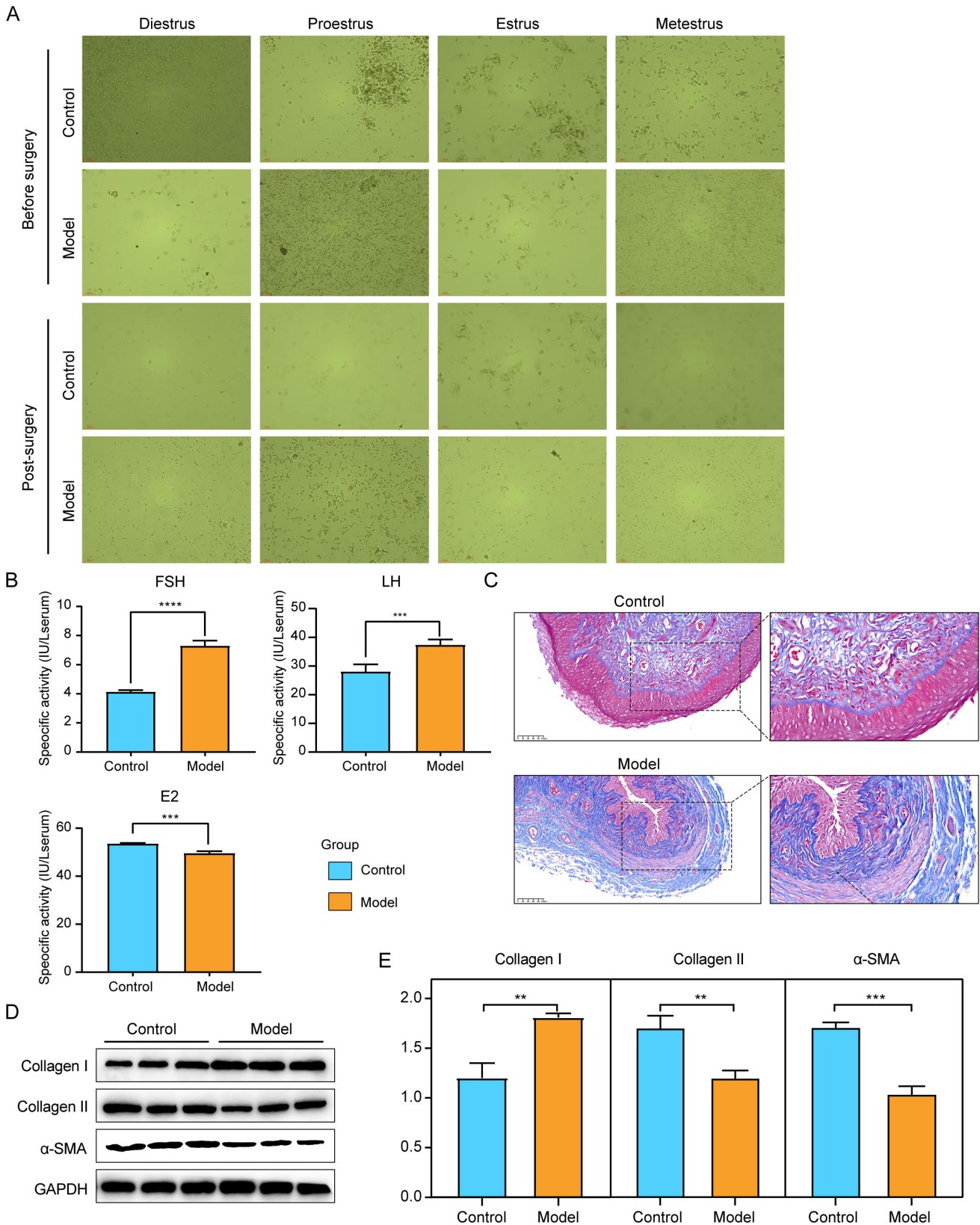

**FIG 1** Establishment of a perimenopausal rat model. (A) The characteristics of cellular changes in vaginal smears of rats during estrous cycle. (B) The levels of FSH, LH, and E2 in different groups were determined by ELISA. (C) Histology examination between control and model groups. (D–E) The protein levels of collagen I, collagen II, and α-SMA in different groups were measured by western blotting. **$P < 0.01$, ***$P < 0.001$, ****$P < 0.0001$.

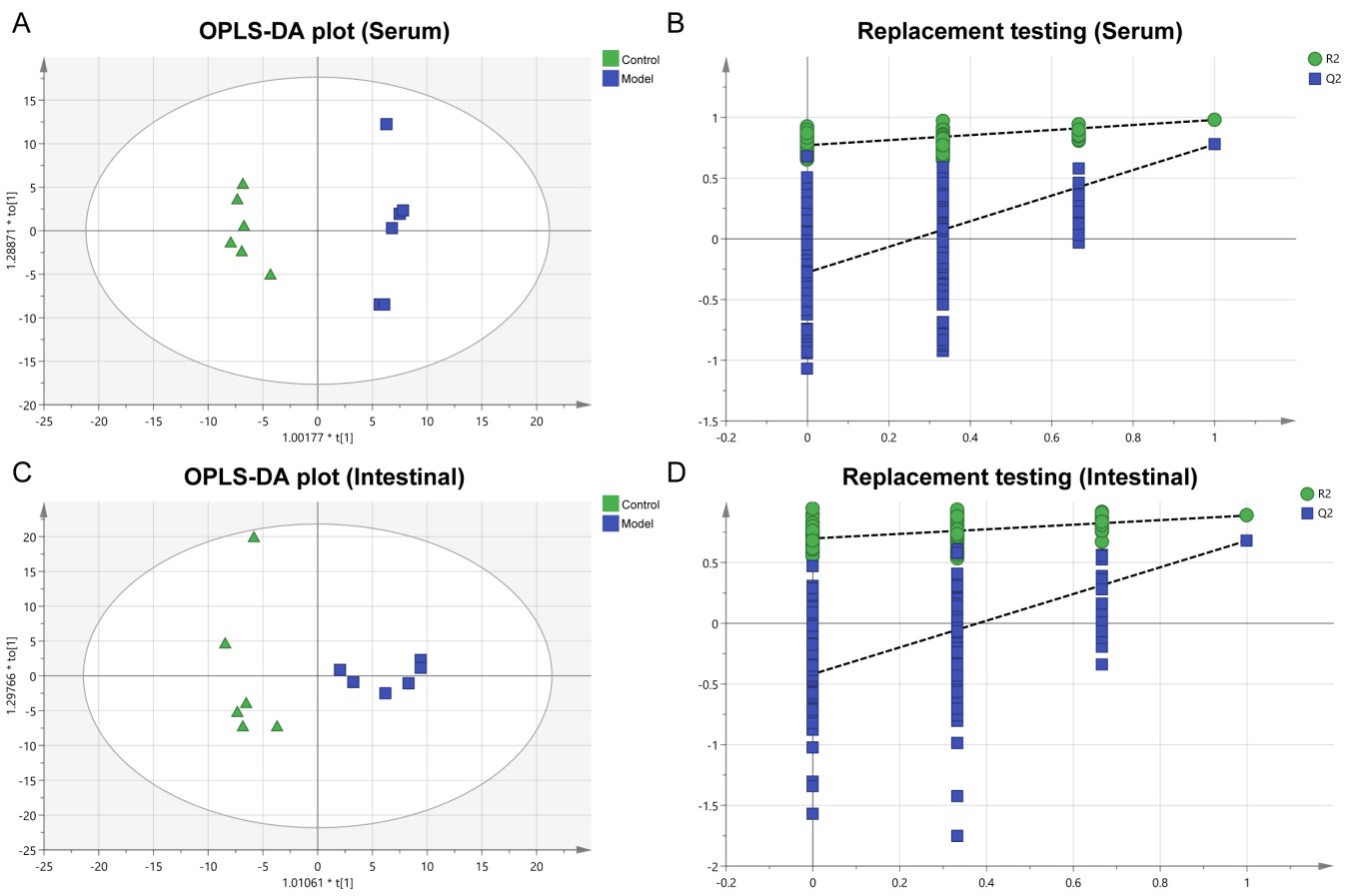

**FIG 2** Multivariate data analysis of serum and intestinal metabolites. (A) OPLS-DA (serum samples). (B) Replacement test (serum samples). (C) OPLS-DA (intestinal samples). (D) Replacement test (intestinal samples).

and derivatives (10, 12.82%), organoheterocyclic compounds (8, 10.26%), not available (6, 7.69%), and nucleosides, nucleotides, and analogs (6, 7.69%) (Fig. 4B). The results of the KEGG pathway enrichment analysis were displayed 20 pathways in model and control *via* adjust *P*-value (FDR < 0.05), metabolites count, and enrichment ratio. In serum samples, the ABC transporters pathway was the most significantly relevant metabolic pathway (Fig. 4C). And the Purine metabolism pathway was the most significantly relevant metabolic pathway in intestinal tissue samples (Fig. 4D).

### Diversity and functional enrichment analysis of gut microbes

The Venn diagram (Fig. 5A) demonstrates that 859 OTUs were shared between the control group and the model group. There were 470 unique OTUs in the model group and 353 unique OTUs in the control group, suggesting significant differences in microbial composition between the two groups. Nonmetric multidimensional scaling analysis and principal coordinate analysis (PCoA) based on the distribution of the OTUs were performed to elucidate the microbiome space of different samples. In the PCoA diagram (Fig. 5B), samples from the Con (red) and test (blue) groups are distinctly separated on

**TABLE 1** OPLS-DA parameter scores[a]

| Sample | $R^2X$ (cum) | $R^2Y$ (cum) | $Q^2$ (cum) |
|---|---|---|---|
| Serum | 0.569 | 0.979 | 0.778 |
| Intestinal | 0.573 | 0.887 | 0.681 |

[a]$R^2X$, the explanation rate of the X matrices; $R^2Y$, the explanation rate of the Y matrices; $Q^2$, the prediction ability; cum, cumulative.

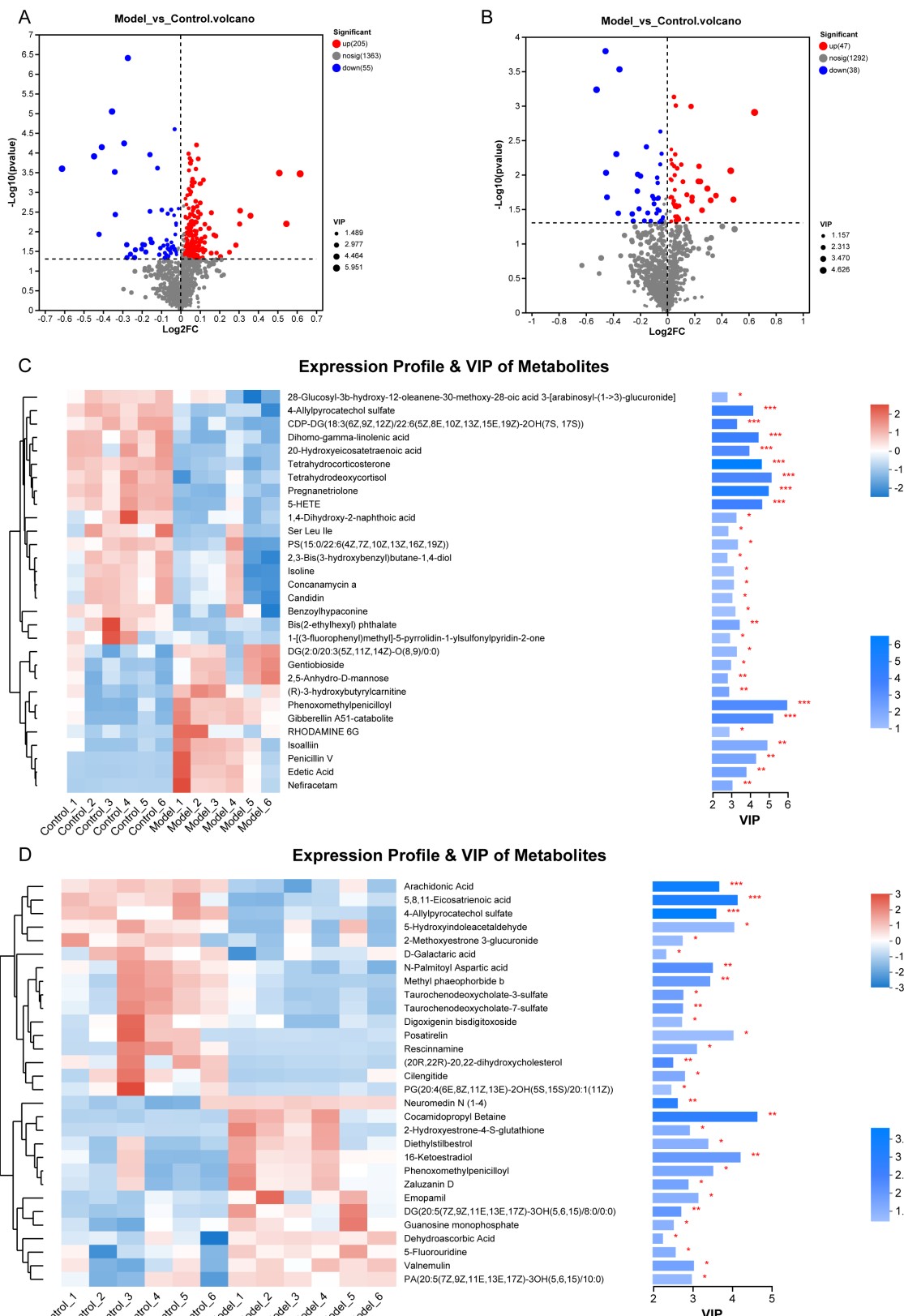

**FIG 3** Differential metabolite expression profiling. (A) The volcano plots at the cation level. (B) The volcano plots at the anion level. The results of thermogram analysis in control and model groups at the cation (C) and the anion (D) level.

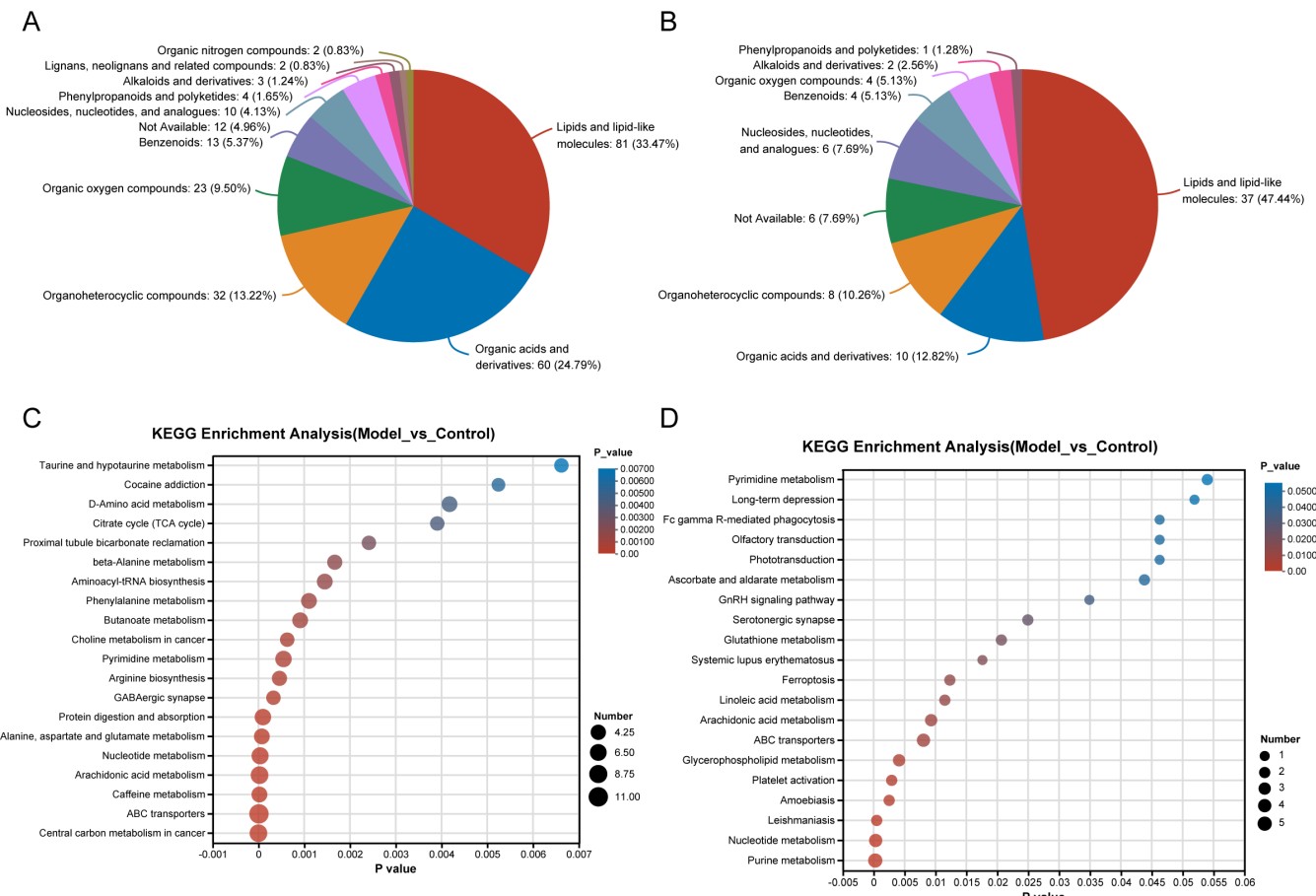

**FIG 4** Metabolite classification and KEGG functional enrichment analysis. Classification of HMDB compounds as metabolites in serum (A) and intestinal tissue (B) samples. KEGG pathway enrichment analysis of differentially expressed metabolites in serum (C) and intestinal tissue (D) samples.

the PC1 axis, and the contribution rate of PC1 to sample separation was 31.24%, which also confirmed a significant difference in the fecal microbial community between the two groups. Utilizing OTU annotations, the relative microbial abundance was calculated across the phylum, class, order, family, and genus levels for each fecal sample. The Wilcoxon rank-sum test was used to analyze the significant differences in microbial composition between groups. FDRs and *q*-values were calculated to adjust *P*-values (27). The average compositions and relative abundances of the microbial community in the two groups at the phylum and genus levels are shown in Fig. 5C and D. At the phylum and genus levels, the top 11 average relative abundance of microbiota were showed (Fig. 5C and D). At the phylum level (Fig. 5C), the cumulative average proportion of *Firmicutes*, *Bacteroidota*, *Actinobacteriota,* and *Campilobacterota* accounted for more than 98% of the two groups. The bar plots revealed that *Deferribacterota* has significant changes between the groups ($P < 0.05$). There were no significant changes between the groups in terms of *Firmicutes* ($P > 0.05$) (Fig. 5E). However, the relative abundance of *Ruminococcus* and *Intestinimonas* in the tests was higher than that in the Controls at the genus level ($P < 0.05$) (Fig. 5F). Functional prediction of straight homologous taxa (COG) of intestinal flora with differences in abundance revealed that these flora had the highest association with Translation, ribosomal structure and biogenesis, amino acid transport and metabolism, followed by transcription and carbohydrate transport and metabolism (Fig. 5G).

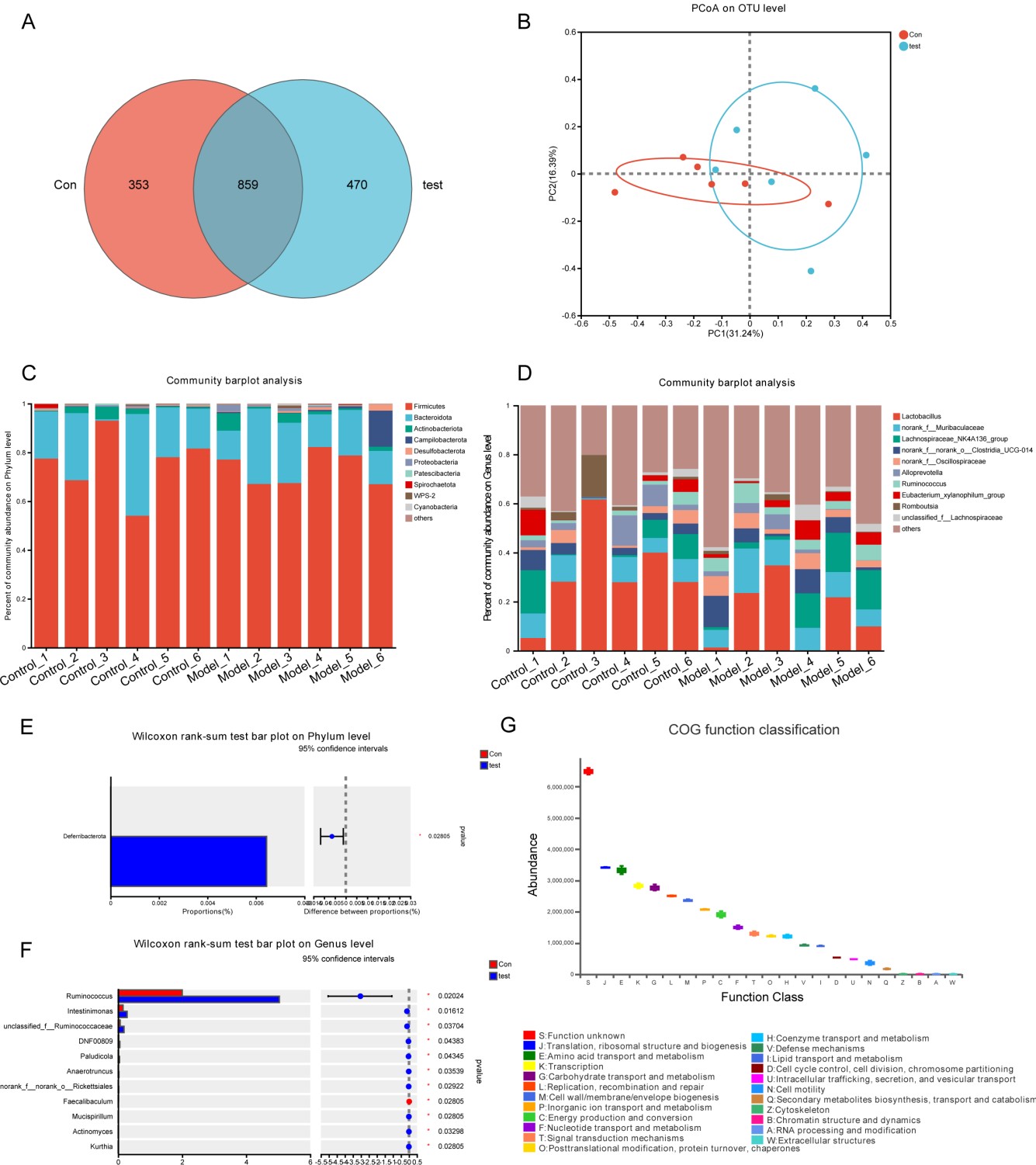

FIG 5 Diversity and functional enrichment analysis of gut microbes. (A) The Venn diagram displays overlap between the two groups. Con, rats with sham operation control groups; test, perimenopausal model groups; OTUs, operational taxonomic units. (B) PCoA of unweighted UniFrac PC1-2 showed that the samples from the Con and test (blue) groups were distinctly separated along the PC1 axis, which means that the overall fecal microbiota compositions were markedly different between Con and tests. Composition of the fecal microbiota at the (C) phylum and (D) genus levels in Con versus test. A Wilcoxon rank-sum test bars plot on phylum level (E) and genus level (F). (G) Functional prediction of straight homologous taxa (COG) of intestinal flora.

## Analysis of the correlation between gut microbes and metabolites

We explored the correlation between differential gut flora and metabolites using Spearman's correlation analysis in serum and intestinal tissue samples (Fig. 6). In serum samples, *Intestinimonas* exhibited the strongest positive correlation with gibberellin A51-catabolite, followed by edetic acid, isoalliin, penicillin V, and phenoxomethylpenicil-loyl; it also showed negative correlations with 1,4-dihydroxy-2-naphthoic acid, 5-HETE, and so on.

*Anaerotruncus* demonstrated the strongest negative correlation with 28-glucosyl-3b-hyfroxy-12-oleanene-30-methoxy-28-oic acid 3-[arabinosyl-(1- > 3)-glucuronide] and positive correlation with edetic acid, followed by penicillin V and phenoxomethylpeni-cilloyl. For additional details on the correlation between gut flora and metabolites, refer to Fig. 6A. In intestinal tissue samples, *Deferribacterota* showed the strongest positive correlation with dehydroascorbic acid, followed by cocamidoprophy betaine, and exhibited negative correlations with N-palmitoyl aspartic acid, arachidonic acid, and so on. *Actinomyces* demonstrated the strongest negative correlations with 5,8,11-eicosa-trienoic acid, followed by 2-methoxyestrone 3-glucuronide. Additional correlation results are displayed in Fig. 6B. Red indicates positive correlation and blue indicates negative correlation. Consequently, the composition of metabolites was associated with variations in gut microbiota across both groups of samples.

## DISCUSSION

Since the metabolic pathways of PMS were still unclear, we embarked on a metabolomics study. In this study, we successfully established a perimenopausal rat model. We then explored the serum and intestinal tissue differential metabolites in rats with PMS *via* a UHPLC-MS/MS-based metabolomics approach. We identified a total of 3,000 metabolites, with 1,623 at the cation level and 1,377 at the anion level. With respect to the key metabolic pathways, a total of 20 pathways were identified by KEGG and MetaboAnalyst. The findings of the present study are useful to further understand the pathophysiologi-cal changes in the perimenopausal state, which can provide clues to explore effective therapies in treating these pathophysiological dysfunctions and contribute to preventing the occurrence of PMS.

In this study, we found a total of 161 HMDB compounds in serum and 78 HMDB compounds in intestinal tissue samples. Moreover, through KEGG analysis, we found that the ABC transporter pathway was the most obvious metabolic pathway in serum samples. In humans, ABC transporters constitute the largest family of transmembrane proteins, and play pivotal roles in cellular homeostasis, including mitochondrial iron regulation, cholesterol metabolism, immune response, and drug sensitivity (28). The dysregulation of these transporters has been associated with the pathogenesis of cystic fibrosis (28, 29). In addition, some studies have reported that ABC transporters are associated with multidrug resistance during the treatment process of uterine cancer (30). Under drug-resistant conditions, the overexpression of these proteins occurs, leading to active efflux of anti-cancer drugs (30). Consequently, resistant cells can evade the cytotoxic effects of drugs and subsequently escape treatment (30). Taken together, these results indicated that the enrichment of ABC transporters in serum samples may be associated with the progression of PMS. Furthermore, in intestinal tissue samples, we found that the purine metabolism pathway was the most significantly distinct metabolic pathway. The purine metabolism pathway can be activated by various factors such as ischemia-induced ATP depletion, consumption of fructose and alcohol, degradation of RNA and DNA due to cell turnover, and intake of a diet rich in purines (31). This metabolic process encompasses *de novo* synthesis, catabolism, and salvage pathways (31). Previous studies have shown that purine metabolism is disturbed by various gynecological diseases, such as ovarian cancer (32), pelvic organ prolapse (33), and endometriosis (34). Herein, purine metabolism disorders in intestinal tissue samples of PMS rats suggest that purine metabolism is associated with the progression of PMS. We

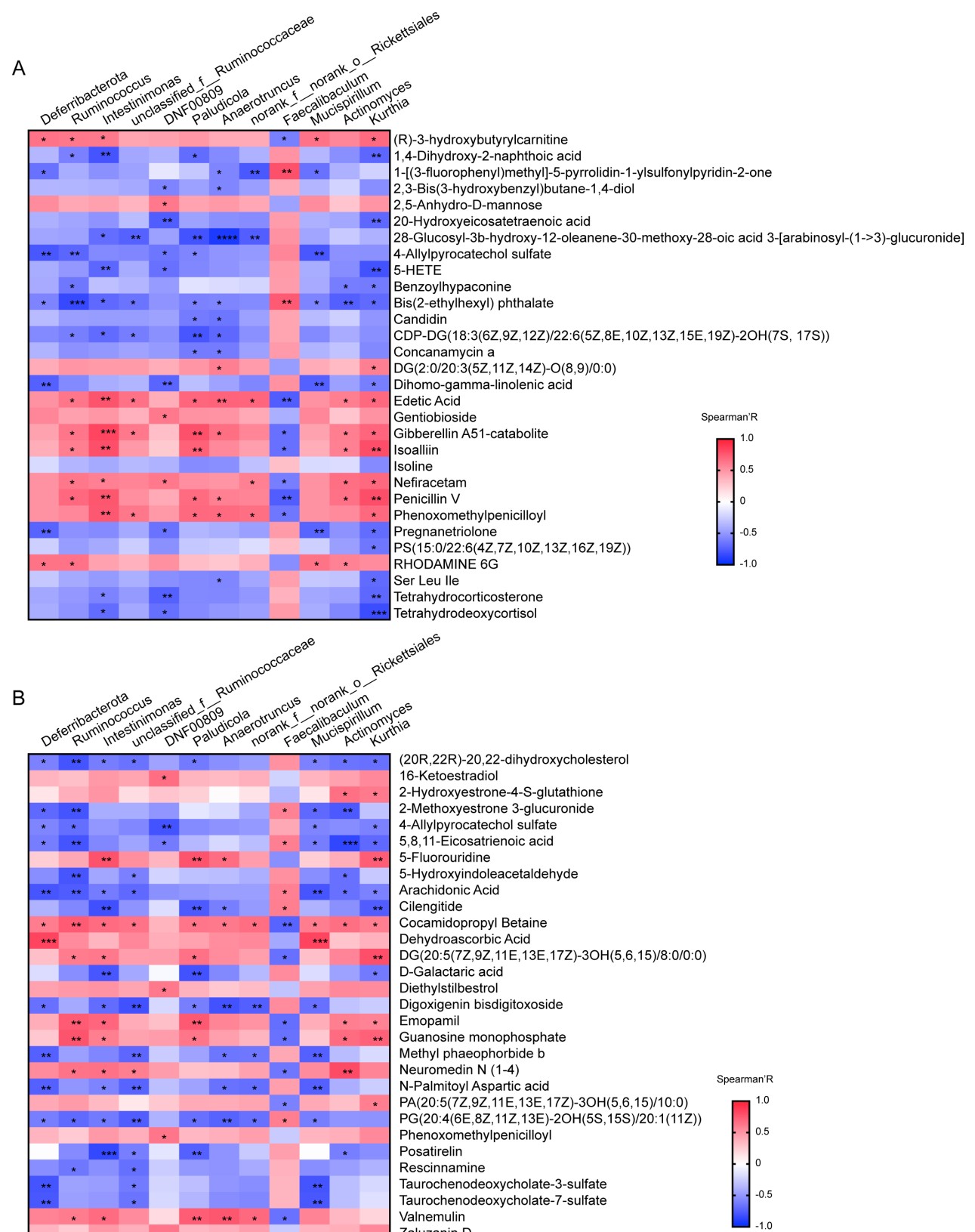

FIG 6  Analysis of the correlation between gut microbes and metabolites. Results of the analysis of differential gut flora in correlation with metabolites in serum samples (A) and intestinal tissue samples (B). Red represents a positive correlation and blue represents a negative correlation. *P < 0.05, **P < 0.005, ***P < 0.0005.

speculated that targeting inhibition of purine metabolism may be effective in inhibiting PMS progression. Because of the explorative nature of the present study, we attempted to find more potential differential metabolites for further verification. The physiological or pathological meanings of the differential metabolites involved in this study still need to be further analyzed and explored.

A growing number of studies showed that gut microbes are closely related to health and disease. And gut microbes are regarded as one of the organs of the human body. The findings showed that the gut microbiome composition was changed in PMS patients: the abundance of 20 species differed significantly between PMS rats and control healthy rats. The *Intestinimonas* are primarily related to inflammation and infections. We found underlying and intriguing relationships between gut microbe composition and function and PMS. Functionally, the PMS is rich in metabolic pathways related to abnormal metabolite transport diseases and purine metabolism disorders, which means that the incidence of gout, anemia, epilepsy, delayed development, obesity, diabetes, and other related diseases may rise during menopause. Current studies also have suggested a potentially strong association between gut microbiota, bone remodeling, and bone metabolic diseases (35). Gut microbiota disorders may cause increased gut permeability and trigger activation of key inflammatory pathways for inducing bone loss in sex steroid-deficient mice (36). Probiotics have shown a positive effect on the management of healthy bone (37). Hence, further comprehensive research is necessary to examine the precise effects of the modified composition of the gut microbiome on the pathophysiology of individuals with PMS.

## Conclusion

In conclusion, our study revealed taxonomic signatures associated with PMS in gut microbes and predicted their function. We also observed the composition of metabolites in the serum and intestinal tissues of PMS model rats, which were linked to variations in their gut microbiota. This suggests that PMS is associated with gut microbiota dysbiosis, specifically a deficiency in the abundance of *Aggregatibacter segnis*, *Bifidobacterium animalis*, and *Acinetobacter guillouiae*, which are related to sex hormone levels. In addition, we identified species with altered abundances and unique functional pathways in PMS individuals. Nevertheless, several limitations to our study should be considered. First, the sample size was limited. Second, further research is needed to determine whether probiotics and fecal transplantation are effective in preventing potential risks in postmenopausal women. Lastly, future multi-omics studies with a larger longitudinal cohort will be required to validate our findings and gain a better understanding of the underlying mechanisms of gut microbiota in PMS. Our study not only provides new insights into the disease mechanisms of PMS but also suggests potential novel therapies to improve the well-being of women after menopause.

## ACKNOWLEDGMENTS

This work was supported by the Natural Science Foundation of Shandong Province (Project No: ZR2023MH380).

Conceptualization, W.W. and X.C.; Methodology, Y.W., J.S., J.W., Z.H., and M.W.; Software, Y.W., J.S., J.W., Z.H., and M.W.; Validation, W.W. and X.C.; Formal Analysis, Y.W., J.S., J.W., Z.H., and M.W.; Investigation, Y.W.; Resources, W.W. and X.C.; Data Curation, W.W. and X.C.; Writing—Original Draft Preparation, Y.W.; Writing—Review & Editing, all authors.; Visualization, Y.W., J.S., J.W., Z.H., and M.W.; Supervision, W.W. and X.C.; Project Administration, W.W. and X.C.; Funding Acquisition, W.W.

## AUTHOR AFFILIATIONS

[1]Department of Gynecology, Tengzhou Central People's Hospital, Tengzhou, Shandong, China

2Translational Pharmaceutical Laboratory, Jining No. 1 People's Hospital, Jining, Shandong, China

3Pelvic Floor Rehabilitation Center, Tengzhou Central People's Hospital, Tengzhou, Shandong, China

4Department of Traditional Chinese Medicine, Tengzhou Central People's Hospital, Tengzhou, Shandong, China

## AUTHOR ORCIDs

Wen Wang  http://orcid.org/0000-0002-6545-2265
Xiujuan Cui  http://orcid.org/0009-0000-5856-9265

## FUNDING

| Funder | Grant(s) | Author(s) |
| --- | --- | --- |
| 山东省科学技术厅 \| Natural Science Foundation of Shandong Province (山东省自然科学基金) | ZR2023MH380 | Wen Wang |

## AUTHOR CONTRIBUTIONS

Yanqiu Wei, Methodology | Juanjuan Shi, Methodology | Jianhua Wang, Methodology | Zongyan Hu, Methodology | Min Wang, Methodology | Wen Wang, Conceptualization, Validation | Xiujuan Cui, Conceptualization, Validation

## DATA AVAILABILITY

The metabolomics data presented in the study are deposited in the OMIX repository, accession number OMIX007622. The 16S rRNA data presented in the study are deposited in the NCBI BioProject repository, accession number PRJNA1141245.

## ETHICS APPROVAL

All experimental procedures conformed to the Guidelines for the Use of Laboratory Animals and were approved by the Ethical Committee for Animal Experimentation at Jining No. 1 People's Hospital (Approval No. JNRM-2023-DW-059).

## ADDITIONAL FILES

The following material is available online.

Open Peer Review

**PEER REVIEW HISTORY (review-history.pdf).** An accounting of the reviewer comments and feedback.

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
