## [Reviewer comments · mSystems]

Integrated analysis of metabolome and microbiome in a rat model of perimenopausal syndrome

Xiujuan Cui, Wen Wang, Yanqiu Wei, Juanjuan Shi, Jianhua Wang, Zongyan Hu, and Min Wang

Corresponding Author(s): Xiujuan Cui, Tengzhou Central People's Hospital

Review Timeline:

Submission Date:	May 6, 2024
Editorial Decision:	May 21, 2024
Revision Received:	July 18, 2024
Editorial Decision:	July 26, 2024
Revision Received:	August 4, 2024
Accepted:	August 13, 2024

Editor: Jack Gilbert

Reviewer(s): Disclosure of reviewer identity is with reference to reviewer comments included in decision letter(s). The following individuals involved in review of your submission have agreed to reveal their identity: Marisol I Dothard (Reviewer #1)

Transaction Report:

DOI: <https://doi.org/10.1128/msystems.00623-24>

Re: mSystems00623-24 (**Integrated analysis of metabolome and microbiome in a rat model of perimenopausal syndrome**)

Dear Dr. Xiujuan Cui:

Revision Guidelines

Sincerely,
Jack Gilbert
Editor
mSystems

Reviewer #1 (Comments for the Author):

The introduction mentions few studies exist in this area but does not mention any other relevant post-menopausal metabolomic studies. Though the majority of other metabolomics studies have focused on plasma, at least one other study has done fecal metabolomics in post-menopausal women but used human samples, and also collected plasma samples. (Lv et al, 2024, Journal of Pharmaceutical and Biomedical Analysis)

Please provide more details for how the OPLS-DA model was verified.

Please fix "more other" in line 387.

The reviewer recommends reconsidering the use of "PMS" in the context of menopause, as this is an acronym that traditionally represents "pre-menstrual syndrome". MS would be more appropriate.

Overall, research is novel and bridges a crucial gap in our understanding of functional microbial changes related to menopause.

The introduction mentions few studies exist in this area but does not mention any other relevant post-menopausal metabolomic studies. Though the majority of other metabolomics studies have focused on plasma, at least one other study has done fecal metabolomics in post-menopausal women but used human samples, and also collected plasma samples. (Lv et al, 2024, Journal of Pharmaceutical and Biomedical Analysis)

Please provide more details for how the OPLS-DA model was verified.

Please fix "more other" in line 387.

The reviewer recommends reconsidering the use of "PMS" in the context of menopause, as this is an acronym that traditionally represents "pre-menstrual syndrome". MS would be more appropriate.

Overall, research is novel and bridges a crucial gap in our understanding of functional microbial changes related to menopause.

Response to Reviewer #1:

The introduction mentions few studies exist in this area but does not mention any other relevant post-menopausal metabolomic studies. Though the majority of other metabolomics studies have focused on plasma, at least one other study has done fecal metabolomics in post-menopausal women but used human samples, and also collected plasma samples. (Lv et al, 2024, Journal of Pharmaceutical and Biomedical Analysis)

Response: Thank you for your insightful comments regarding the context of existing literature in post-menopausal metabolomic studies. We appreciate the reference to Lv et al. (2024) including those involving fecal metabolomics, enriches the background and underscores the significance of our work. Compared with Lv et al.'s study, there are two main differences:

(1) Differences in Experimental Design with Lv et al. (2024).

While both studies incorporate fecal samples, the experimental designs differ. Lv et al. focused on the impact of a traditional Chinese medicine formula (Geng-Nian-Shu, GNS) on the metabolomic profiles of blood and fecal samples in a perimenopausal rat model. In contrast, our study investigates the changes in blood and intestinal tissue metabolomics in normal versus perimenopausal rats. We also conduct 16SrRNA sequencing to analyze differences in microbial communities in the cecal contents (fecal samples) between normal and perimenopausal rats. This dual approach allows us to explore both the metabolic and microbial changes associated with perimenopause more comprehensively.

(2) Differences in Key Findings with Lv et al. (2024).

Lv et al. link metabolic changes during the perimenopausal syndrome (PMS) with ferroptosis, suggesting that GNS might regulate ferroptosis by modulating the metabolic pathways of organic acids and their derivatives, as well as lipids and lipid-like molecules. Our findings, on the other hand, reveal significant changes in the composition of metabolites in serum and intestinal tissues of PMS model rats, which are associated with shifts in their gut microbiota. These changes suggest that premenstrual syndrome is linked to gut microbiota dysbiosis, especially marked by a deficiency in the abundance of *Aggregatibacter segnis*, *Bifidobacterium animalis*, and *Acinetobacter guillouiae*, which are related to hormone levels. Our study not only provides new insights into the disease mechanisms of PMS, but also suggests potential new therapies to improve the gut microbiome of postmenopausal women.

In the revised version, we have added a discussion about utilizing metabolome and microbiome to study perimenopausal syndrome (PMS) in the "Introduction" section, and highlighted the metabolome and microbiome of PMS to improve our understanding the comprehensive characterizations of PMS. The details were shown as follows **(Lines 98-102, page 4)**.

Lv et al. link metabolic changes during the perimenopausal syndrome (PMS) with ferroptosis, suggesting that GNS might regulate ferroptosis by modulating the metabolic pathways of organic acids and their derivatives, as well as lipids and lipid-like molecules. However, there is still a lack of research on comprehensive analysis of perimenopausal metabolism and 16s rRNA.

Please provide more details for how the OPLS-DA model was verified.

Response: Thank you for your insightful query regarding the verification of the OPLS-DA model used in our study. To ensure the robustness and reliability of our model, we implemented several verification steps as follows:

(1) Model Validation: The OPLS-DA model was rigorously validated using a permutation test, with 200 permutations, to prevent model overfitting. This test helps in assessing the significance of the predictive capability of the model compared to a random classification.

(2) Cross-Validation: We employed sevenfold cross-validation to evaluate the predictive accuracy of the model. This method partitions the data into seven subsets, uses six for training, and one for testing, iteratively, to ensure that each data point is used for both training and validation.

(3) Performance Metrics: The quality of the model was further assessed using the R^2Y (cumulative modeled variation in Y) and Q^2 (predictive ability) values. A Q^2 value greater than 0.5 indicates a model with excellent predictive reliability. The parameter scores of OPLS-DA were shown in **Table 1**. In serum and intestinal samples, the proportion of variance explained by the OPLS-DA model (R^2Y) was 97.9%, and 88.7%. The cumulative values of Q^2 in the serum and intestinal samples (0.778, 0.681) were all above 0.50, which illustrated the stability and reliability of the model.

Table 1. OPLS-DA parameter scores.

Sample	R2X(cum)	R2Y(cum)	Q2(cum)
serum	0.569	0.979	0.778
intestinal	0.573	0.887	0.681

R^2X : the explanation rate of the X matrices; R^2Y : the explanation rate of the Y matrices; Q^2 : the prediction ability

(4) Variable Importance in Projection (VIP) Scores: Variables with VIP scores greater than 1.0 were considered significant contributors to the model, highlighting their importance in distinguishing between the groups studied.

These validation steps confer robustness to our OPLS-DA model, ensuring that the metabolomic differences observed are statistically significant and not due to random chance. This rigorous validation approach underscores the reliability of the findings reported in our study.

In the revised version, we have added this section in the “**Introduction**” part of the revised manuscript with red text (**Lines 63-66, page 3**).

Please fix "more other" in line 387.

Response: Thank you for your careful review and constructive suggestions regarding our manuscript,

and we are very sorry for our negligence of syntax error. In the revised version, “more other” was changed as “additional” (**Line 386, page 13**).

Furthermore, the revised manuscript was also edited by professional native speakers. All changes were highlighted with red text in the revised manuscript. Please see the details in the revised manuscript.

The reviewer recommends reconsidering the use of "PMS" in the context of menopause, as this is an acronym that traditionally represents "pre-menstrual syndrome". MS would be more appropriate. Overall, research is novel and bridges a crucial gap in our understanding of functional microbial changes related to menopause.

Response: We appreciate the constructive comments and professional suggestions.

I acknowledge that "PMS" is traditionally associated with "pre-menstrual syndrome." However, upon reviewing relevant literature, I found that there are instances where "PMS" has been used to describe "perimenopausal syndrome" [1-3], Additionally, "MS" typically refers to "menopausal syndrome" in the literature [4], which represents a distinct phase compared to the perimenopausal period.

Given this context, I have chosen to use "PMS" to specifically denote "perimenopausal syndrome" in this study to maintain clarity between these different phases of women's reproductive aging. However, to avoid confusion, I will ensure that the full term "perimenopausal syndrome" is clearly defined at the first use of the acronym in the manuscript (**Line 38, page 2**), and clarify the distinction between "perimenopausal" and "menopausal" phases to help the readers understand the specific focus of our study (**Line 63-66, page 3**).

Reference:

- 1 Junjie, L. *et al.* An Effective Treatment of Perimenopausal Syndrome by Combining Two Traditional Prescriptions of Chinese Botanical Drugs. *Frontiers in Pharmacology*, doi:10.3389/fphar.2021.744409 (2021).
- 2 Xinyan, L. *et al.* Non-targeted metabolomics strategy reveals the role of Geng-Nian-Shu in regulating ferroptosis in perimenopausal syndrome. *Journal of Pharmaceutical and Biomedical Analysis*, doi:10.1016/j.jpba.2024.115980 (2024).
- 3 Beibei, X. *et al.* Application of multivariate statistical analysis and network pharmacology to explore the mechanism of Danggui Liu Huang Tang in treating perimenopausal syndrome. *Journal of Ethnopharmacology*, doi:10.1016/j.jep.2021.114543 (2022).
- 4 Ming, T. *et al.* Study on the mechanism of Baihe Dihuang decoction in treating menopausal syndrome based on network pharmacology. *Medicine*, doi:10.1097/md.00000000000033189 (2023).

Re: mSystems00623-24R1 (**Integrated analysis of metabolome and microbiome in a rat model of perimenopausal syndrome**)

Dear Dr. Xiujuan Cui:

We are ready to accept your article based on the modifications, but the sequence data and other data need to be made available as per our data policy. <https://journals.asm.org/open-data-policy>

Revision Guidelines

Sincerely,
Jack Gilbert
Editor
mSystems

Editor:

We are ready to accept your article based on the modifications, but the sequence data and other data need to be made available as per our data policy.

<https://journals.asm.org/open-data-policy>

Response:

Thanks very much for your kind work and consideration on publication of our paper. Sequence data and other data have been uploaded to the appropriate database in accordance with your data policy, and we modified the “Data availability Statement” in the manuscript as required.

Re: mSystems00623-24R2 (**Integrated analysis of metabolome and microbiome in a rat model of perimenopausal syndrome**)

Dear Dr. Xiujuan Cui:

Your manuscript has been accepted, and I am forwarding it to the ASM production staff for publication. Your paper will first be checked to make sure all elements meet the technical requirements. ASM staff will contact you if anything needs to be revised before copyediting and production can begin. Otherwise, you will be notified when your proofs are ready to be viewed.

Sincerely,
Jack Gilbert